# Human Decision-Making under Limited Time

**Pedro A. Ortega**
Department of Psychology
University of Pennsylvania
Philadelphia, PA 19104
ope@seas.upenn.edu

**Alan A. Stocker**
Department of Psychology
University of Pennsylvania
Philadelphia, PA 19014
astocker@sas.upenn.edu

## Abstract

Subjective expected utility theory assumes that decision-makers possess unlimited computational resources to reason about their choices; however, virtually all decisions in everyday life are made under resource constraints—i.e. decision-makers are bounded in their rationality. Here we experimentally tested the predictions made by a formalization of bounded rationality based on ideas from statistical mechanics and information-theory. We systematically tested human subjects in their ability to solve combinatorial puzzles under different time limitations. We found that our bounded-rational model accounts well for the data. The decomposition of the fitted model parameter into the subjects' expected utility function and resource parameter provide interesting insight into the subjects' information capacity limits. Our results confirm that humans gradually fall back on their learned prior choice patterns when confronted with increasing resource limitations.

## 1 Introduction

Human decision-making is not perfectly rational. Most of our choices are constrained by many factors such as perceptual ambiguity, time, lack of knowledge, or computational effort [6]. Classical theories of rational choice do not apply in such cases because they ignore information-processing resources, assuming that decision-makers always pick the optimal choice [10]. However, it is well known that human choice patterns deviate qualitatively from the perfectly rational ideal with increasing resource limitations.

It has been suggested that such limitations in decision-making can be formalized using ideas from statistical mechanics [9] and information theory [16]. These frameworks propose that decision-makers act *as if* their choice probabilities were an optimal compromise between maximizing the expected utility and minimizing the KL-divergence from a set of prior choice probabilities, where the trade-off is determined by the amount of available resources. This optimization scheme reduces the decision-making problem to the *inference* of the optimal choice from a stimulus, where the likelihood function results from a combination of the decision-maker's subjective preferences and the resource limitations.

The aim of this paper is to systematically validate the model of bounded-rational decision-making on human choice data. We conducted an experiment in which subjects had to solve a sequence of combinatorial puzzles under time pressure. By manipulating the allotted time for solving each puzzle, we were able to record choice data under different resource conditions. We then fit the bounded-rational choice model to the dataset, obtaining a decomposition of the choice probabilities in terms of a resource parameter and a set of stimulus-dependent utility functions. Our results show that the model captures very well the gradual shifts due to increasing time constraints that are present in the subjects' empirical choice patterns.

## 2  A Probabilistic Model of Bounded-Rational Choices

We model a bounded-rational decision maker as an expected utility maximizer that is subject to information constraints. Formally, let $\mathcal{X}$ and $\mathcal{Y}$ be two finite sets, the former corresponding to a *set of stimuli* and the latter to a *set of choices*; and let $P(y)$ be a prior distribution over optimal choices $y \in \mathcal{Y}$ that the decision-maker may have learned from experience. When presented with a stimulus $x \in \mathcal{X}$, a bounded-rational decision-maker transforms the prior choice probabilities $P(y)$ into posterior choice probabilities $P(y|x)$ and then generates a choice according to $P(y|x)$.

This transformation is modeled as the optimization of a regularized expected utility known as the *free energy functional*:

$$F\big[Q(y|x)\big] := \underbrace{\sum_y Q(y|x) U_x(y)}_{\text{Expected Utility}} - \underbrace{\frac{1}{\beta} \sum_y Q(y|x) \log \frac{Q(y|x)}{P(y)}}_{\text{Regularization}}, \tag{1}$$

where the posterior is defined as the maximizer $P(y|x) := \arg\max_{Q(y|x)} F[Q(y|x)]$. Crucially, the optimization is determined by two factors. The first is the decision-maker's *subjective utility function* $U_x : \mathcal{Y} \to \mathbb{R}$ encoding the desirability of a choice $y$ given a stimulus $x$. The second is the *inverse temperature* $\beta$, which determines the resources of deliberation available for the decision-task[1], but which are neither known to, nor controllable by the decision-maker. The resulting posterior has an analytical expression given by the Gibbs distribution

$$P(y|x) = \frac{1}{Z_\beta(x)} P(y) \exp\big\{\beta U_x(y)\big\}, \tag{2}$$

where $Z_\beta(x)$ is a normalizing constant [9]. The expression (2) highlights a connection to inference: bounded-rational decisions can also be computed via Bayes' rule in which the likelihood is determined by $\beta$ and $U_x$ as follows:

$$P(y|x) = \frac{P(y)P(x|y)}{\sum_{y'} P(y')P(x|y')}, \quad \text{hence} \quad P(x|y) \propto \exp\big\{\beta U_x(y)\big\}. \tag{3}$$

The objective function (1) can be motivated as a trade-off between maximizing expected utility and minimizing information cost [9, 16]. Near-zero values of $\beta$, which correspond to heavily-regularized decisions, yield posterior choice probabilities that are similar to the prior. Conversely, with growing values of $\beta$, the posterior choice probabilities approach the perfectly-rational limit.

**Connection to regret.**  Bounded-rational decision-making is related to *regret theory* [2, 4, 8]. To see this, define the *certainty-equivalent* as the maximum attainable value for (1):

$$U_x^* := \max_{Q(y|x)} \Big\{ F\big[Q(y|x)\big] \Big\} = \frac{1}{\beta} \log Z_\beta(x). \tag{4}$$

The certainty-equivalent quantifies the *net worth* of the stimulus $x$ prior to making a choice. The decision process treats (4) as a reference utility used in the assessment of the alternatives. Specifically, the modulation of any choice is obtained by measuring up the utility against the certainty-equivalent:

$$\underbrace{\log \frac{P(y|x)}{P(y)}}_{\text{Change of } y} = -\beta \Big[\underbrace{U_x^* - U_x(y)}_{\text{Regret of } y}\Big]. \tag{5}$$

Accordingly, the difference in log-probability is proportional to the negative regret [3]. The decision-maker's utility function specifies a direction of change relative to the certainty-equivalent, whereas the strength of the modulation is determined by the inverse temperature.

# 3 Experimental Methods

We conducted a choice experiment where subjects had to solve puzzles under time pressure. Each puzzle consisted of Boolean formula in conjunctive normal form (CNF) that was disguised as an arrangement of circular patterns (see Fig. 1). The task was to find a truth assignment that satisfied the formula. Subjects could pick an assignment by setting the colors of a central pattern highlighted in gray. Formally, the puzzles and the assignments corresponded to the stimuli $x \in \mathcal{X}$ and the choices $y \in \mathcal{Y}$ respectively, and the duration of the puzzle was the resource parameter that we controlled (see equation 1).

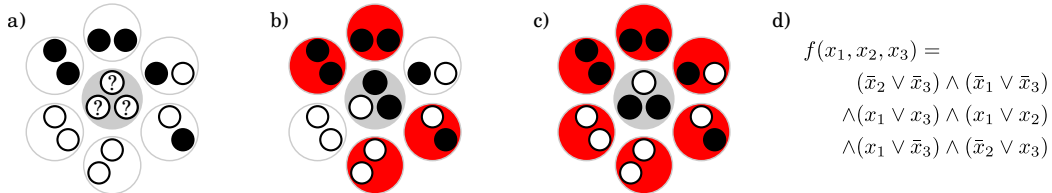

Figure 1: *Example puzzle.* a) Each puzzle is a set of six circularly arranged patches containing patterns of black (●) and white circles (○). In each trial, the positions of the patches were randomly assigned to one of the six possible locations. Subjects had to choose the three center colors such that there was at least one (color and position) match for each patch. For instance, the choice in (b) only matches four out of six patches (in red), while (c) solves the puzzle. The puzzle is a visualization of the Boolean formula in (d).

We restricted our puzzles to a set of five CNF formulas having 6 clauses, 2 literals per clause, and 3 variables. Subjects were trained only on the first four puzzles, whereas the last one was used as a control puzzle during the test phase. All the chosen puzzles had a single solution out of the $2^3 = 8$ possible assignments.

We chose CNF formulas because they provide a general[2] and flexible platform for testing decision-making behavior. Crucially, unlike in an estimation task, finding the relation between a stimulus and a choice is non-trivial and requires solving a computational problem.

## 3.1 Data Collection

Two symmetric versions of the experiment were conducted on Amazon Mechanical Turk. For each, we collected choice data from 15 anonymized participants living in the United States, totaling 30 subjects. Subjects were paid 10 dollars for completing the experiment. The typical runtime of the experiment ranged between 50 and 130 minutes.

For each subject, we recorded a sequence of 90 training and 285 test trials. The puzzles were displayed throughout the whole trial, during which the subjects could modify their choice at will. The training trials allowed subjects to familiarize themselves with the task and the stimuli, whereas the test trials measured their adapted choice behavior as a function of the stimulus and the task duration. Training trials were presented in blocks of 18 for a long, fixed duration; the test trials, which were of variable duration, were presented in blocks of 19 (18 regular + 1 control trial). To avoid the collection of poor quality data, subjects had to repeat a block if they failed more than 6 trials within the same block, thereby setting a performance threshold that was well above chance level. Participants could initiate a block whenever they felt ready to proceed. Within a block, the inter-trial durations were drawn uniformly between 0.5 and 1.5s.

Each trial consisted of one puzzle that had to be solved within a limited time. Training trials lasted 10s each, while test trials had durations of 1.25, 2.5, and 5s. Apart from a visual cue shown 1s before the end of each trial, there was no explicit feedback communicating the trial length. Therefore, subjects did not know the duration of individual test trials beforehand and thus *could not use this information in their solution strategy*. A trial was considered successful only if all the clauses of the puzzle were satisfied.

# 4 Analysis

The recorded data $\mathcal{D}$ consists of a set of tuples $(x, r, y)$, where $x \in \mathcal{X}$ is a stimulus, $r \in \mathcal{R}$ is a resource parameter (i.e. duration), and $y \in \mathcal{Y}$ a choice. In order to analyze the data, we made the following assumptions:

1. *Transient regime*: During the training trials, the subjects converged to a set of subjective preferences over the choices which depended only on the stimuli.

2. *Permanent regime*: During the test trials, subjects did not significantly change the preferences that they learned during the training trials. Specifically, choices in the same stimulus-duration group were i.i.d. throughout the test phase.

3. *Negligible noise*: We assumed that the operation of the input device and the cue signaling the imminent end of the trial did not have a significant impact on the distribution over choices.

Our analysis only focused only the test trials. Let $P(x, r, y)$ denote the empirical probabilities[3] of the tuples $(x, r, y)$ estimated from the data. From these, we derived the probability distribution $P(x, r)$ over the stimulus-resource context, the prior $P(y)$ over choices, and the posterior $P(y|x, r)$ over choices given the context through marginalization and conditioning.

## 4.1 Inferring Preferences

By fitting the model, we decomposed the choice probabilities into: (a) an inverse temperature function $\beta : \mathcal{R} \to \mathbb{R}$; and (b) a set of subjective utility functions $U_x : \mathcal{Y} \to \mathbb{R}$, one for each stimulus $x$. We assumed that the sets $\mathcal{X}$, $\mathcal{R}$, and $\mathcal{Y}$ were finite, and we used vector representations for $\beta$ and the $U_x$. To perform the decomposition, we minimized the average Kullback-Leibler divergence

$$J = \sum_{x,r} P(x, r) \left[ \sum_y P(y|x, r) \log \frac{P(y|x, r)}{Q(y|x, r)} \right], \tag{6}$$

w.r.t. the inverse temperatures $\beta(r)$ and the utilities $U_x(y)$ through the probabilities $Q(y|x, r)$ of the choice $y$ given the context $(x, r)$ as derived from the Gibbs distribution

$$Q(y|x, r) = \frac{1}{Z_\beta} P(y) \exp \left\{ \beta(r) U_x(y) \right\}, \tag{7}$$

where $Z_\beta$ is the normalizing constant. We used the objective function (6) because it is the Bregman divergence over the simplex of choice probabilities [1]. Thus, by minimizing the objective function (6) we were seeking a decomposition such that the Shannon information contents of $P(y|x, r)$ and $Q(y|x, r)$ were matched against each other in expectation.

We minimized (6) using gradient descent. For this, we first rewrote (6) as

$$J = \sum_{x,\beta,y} P(x, r, y) \left\{ \log \frac{P(y|x, r)}{P(y)} - \beta(r) U_x(y) + \log Z_\beta \right\}$$

to expose the coordinates of the exponential manifold and then calculated the gradient. The partial derivatives of $J$ w.r.t. $\beta(r)$ and $U_x(y)$ are equal to

$$\frac{\partial J}{\partial \beta(r)} = \sum_{x,y} P(x, r) \sum_y \Big[ Q(y|x, r) - P(y|x, r) \Big] U_x(y) \tag{8}$$

$$\text{and} \quad \frac{\partial J}{\partial U_x(y)} = \sum_{x,y} P(x, r) \Big[ Q(y|x, r) - P(y|x, r) \Big] \beta(r) \tag{9}$$

respectively. The Gibbs distribution (7) admits an infinite number of decompositions, and therefore we had to fix the scaling factor and the offset to obtain a unique solution. The scale was set by clamping the value of $\beta(r_0) = \beta_0$ for an arbitrarily chosen resource parameter $r_0 \in \mathcal{R}$; we used

$\beta(r_0) = 1$ for $r_0 = 1$s. The offset was fixed by normalizing the utilities. A simple way to achieve this is by subtracting the certainty-equivalent from the utilities, i.e. for all $(x, y)$,

$$U_x(y) \leftarrow U_x(y) - \frac{1}{\beta(r_0)} \log \sum_y P(y) \exp\Big\{\beta(r_0)U_x(y)\Big\}. \tag{10}$$

Utilities normalized in this way are proportional to the negative regret (see Section 2) and thus have an intuitive interpretation as modulators of change of the choice distribution.

The resulting decomposition algorithm repeats the following two steps until convergence: first it updates the inverse temperature and utility functions using gradient descent, i.e.

$$\beta(r) \longleftarrow \beta(r) - \eta_t \frac{\partial J}{\partial \beta(r)} \quad \text{and} \quad U_x(y) \longleftarrow U_x(y) - \eta_t \frac{\partial J}{\partial U_x(y)} \tag{11}$$

for all $(r, x, y) \in \mathcal{R} \times \mathcal{X} \times \mathcal{Y}$; and seconds it projects the parameters back onto a standard submanifold by setting $r = r_0$ and normalizing the utilities in each iteration using (10). For the learning rate $\eta_t > 0$, we choose a simple schedule that satisfied the Robbins-Monro conditions $\sum_t \eta_t = \infty$ and $\sum_t \eta_t^2 < \infty$.

### 4.2 Expected Utility and Decision Bandwidth

The inferred model is useful for investigating the decision-maker's performance under different settings of the resource parameter—in particular, to determine the asymptotic performance limits. Two quantities are of special interest: the *expected utility* averaged over the stimuli and the *mutual information* between the stimulus and the choice, both as functions of the inverse temperature $\beta$. Given $\beta$, we define these quantities as

$$EU_\beta := \sum_{x,y} P(x)Q_\beta(y|x)U_x(y) \quad \text{and} \quad I_\beta := \sum_{x,y} P(x)Q_\beta(y|x) \log \frac{Q_\beta(y|x)}{Q_\beta(y)} \tag{12}$$

respectively. Both definitions are based on the joint distribution $P(x)Q_\beta(y|x)$ in which $Q_\beta(y|x) \propto P(y)\exp\{\beta U_x(x)\}$ is the Gibbs distribution derived from the prior $P(y)$ and the utility functions $U_x(y)$. The marginal over choices is given by $Q_\beta(y) = \sum_x P(x)Q_\beta(y|x)$. The mutual information $I_\beta$ is a measure of the decision bandwidth, because it quantifies the average amount of information that the subject has to extract from the stimulus in order to produce the choice.

## 5 Results

### 5.1 Decomposition into prior, utility, and inverse temperature

For each one of the 30 subjects, we first calculated the empirical choice probabilities and then estimated their decomposition into an inverse temperature $\beta$ and utility functions $U_x$ using the procedure detailed in the previous section. The mean error of the fit was very low ($0.0347 \pm 0.0024$ bits), implying that the choice probabilities are well explained by the model. As an example, Fig. 2 shows the decomposition for subject 1 (error $0.0469$ bits, $83\%$ percentile rank) along with a comparison between the empirical posterior and the model posterior calculated from the inferred components using equation (7). As durations become longer and $\beta$ increases, the model captures the gradual shift from the prior towards the optimal choice distribution.

As seen in Fig. 3, the resulting decomposition is stable and shows little variability across subjects. The stimuli of version B of the experiment differed from version A only in that they were color-inverted, leading to mirror-symmetric decompositions of the prior and the utility functions. The results suggest the following trends:

- *Prior:* Compared to the true distribution over solutions, subjects tended to concentrate their choices slightly more on the most frequent optimal solution (i.e. either $y = 2$ or $y = 7$ for version A or B respectively) and on the all-black or all-white solution (either $y = 1$ or $y = 8$).

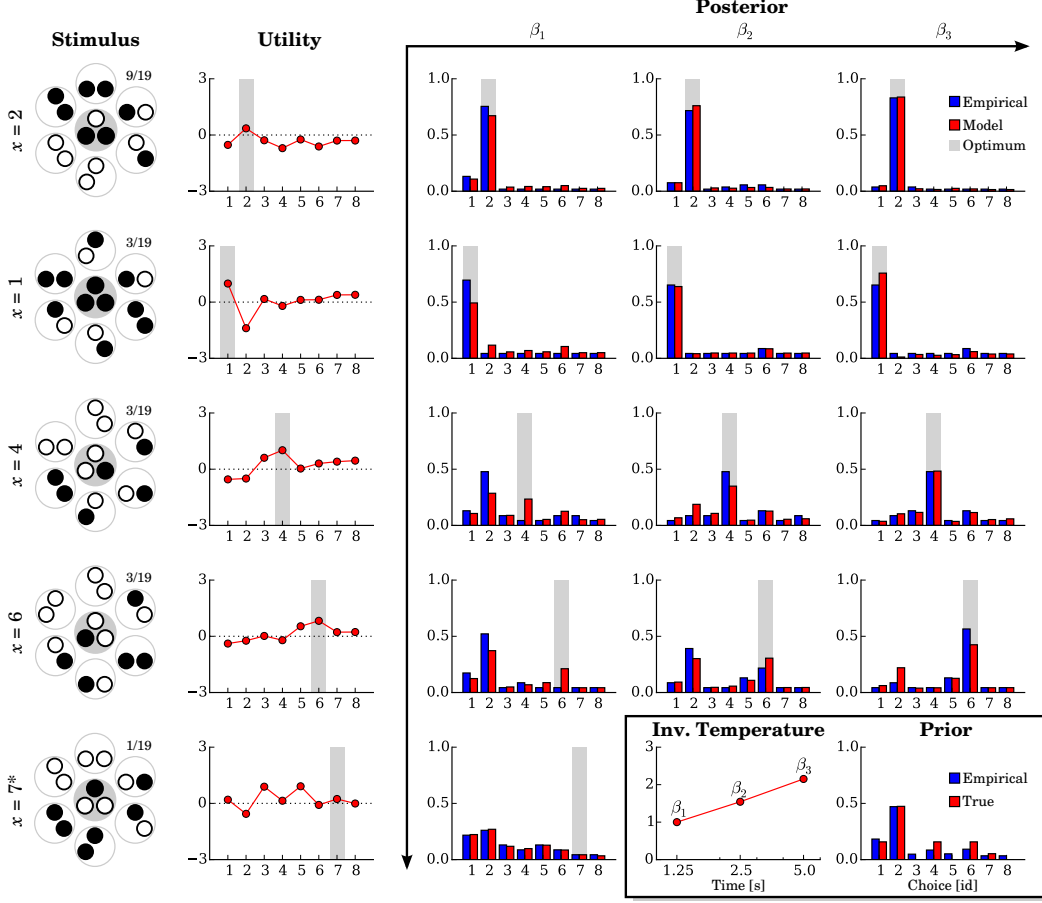

Figure 2: *Decomposition of subject 1's posterior choice probabilities.* Each row corresponds to a different puzzle. The left column shows each puzzle's stimulus and optimal choice. The posterior distributions $P(y|x,\beta)$ were decomposed into a prior $P(y)$; a set of time-dependent inverse temperatures $\beta_r$; and a set of stimulus-dependent utility functions $U_x$ over choices, normalized relative to the certainty-equivalent (10). The plots compare the subject's empirical frequencies against the model fit (in the *posterior* plots) or against the true optimal choice probabilities (in the *prior* plot). The stimuli are shown on the left (more specifically, one out of the 6! arrangement of patches) along with their probability. Note that the untrained stimulus $x = 7$ is the color-inverse of $x = 2$.

- *Inverse temperature:* The inverse temperature increases monotonically with longer durations, and the dependency is approximately linear in log-time (Fig. 2 and 3).

- *Utility functions:* In the case of the stimuli that subjects were trained in (namely, $x \in \{1, 2, 4, 6\}$), the maximum subjective utility coincides with the solution of the puzzle. Notice that some choices are enhanced while others are suppressed according to their subjective utility function. Especially the choice for the most frequent stimulus ($x = 2$) is suppressed when it is suboptimal. In the case of the untrained stimulus ($x = 7$), the utility function is comparatively flat and variable across subjects.

Finally, as a comparison, we also computed the decomposition assuming a *Softmax function* (or *Boltzmann distribution*):

$$Q(y|x,r) = \frac{1}{Z_\beta} \exp\Big\{\beta(r) U_x(y)\Big\}. \tag{13}$$

The mean error of the resulting fit was significantly worse (error $0.0498 \pm 0.0032$ bits) than the one based on (7), implying that the inclusion of the prior choice probabilities $P(y)$ improves the explanation of the choice data.

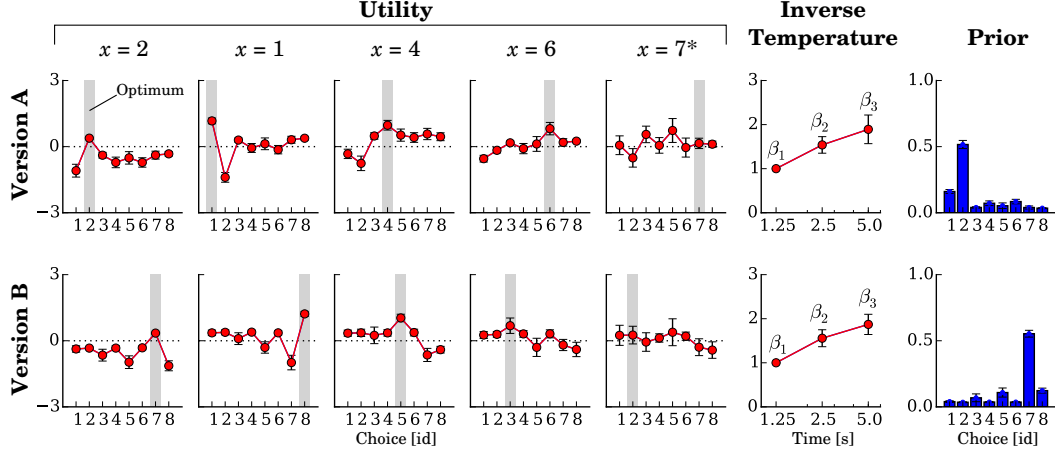

Figure 3: *Summary of inferred preferences across all subjects*. The two rows depict the results for the two versions of the experiment, each one averaged over 15 subjects. The stimuli of both versions are the same but with their colors inverted, resulting in a mirror symmetry along the vertical axis. The figure shows the inferred utility functions (normalized to the certainty-equivalent); the inverse temperatures; and the prior over choices. Optimal choices are highlighted in gray. Error bars denote one standard deviation.

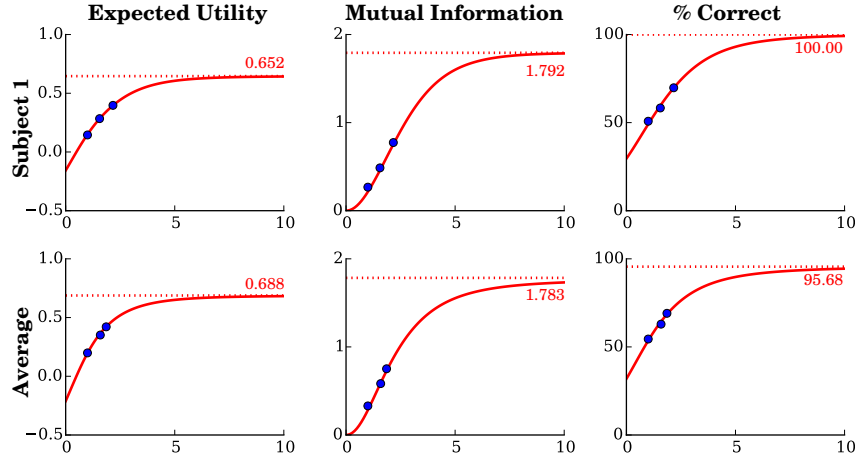

Figure 4: *Extrapolation of the performance measures*. The panels show the expected utility $EU_\beta$, the mutual information $I_\beta$, and the expected percentage of correct choices as a function of the inverse temperature $\beta$. The top and bottom rows correspond to subject 1 and the averaged subjects respectively. Each plot shows the performance measure obtained from the empirical choice probabilities (blue markers) and the choice probabilities derived from the model (red curve) together with the maximum attainable value (dotted red).

## 5.2 Extrapolation of performance measures

We calculated the expected utility and the mutual information as a function of the inverse temperature using (12). The resulting curves for subject 1 and the average subject are shown in Fig. 4 together with the predicted percentage of correct choices. All the curves are monotonically increasing and upper bounded. The expected utility and the percentage of correct choices are concave in the inverse temperature, indicating marginally diminishing returns with longer durations. Similarly, the mutual information approaches asymptotically the upper bound set by the stimulus entropy $H(X) \approx 1.792$ bits (excluding the untrained stimulus).

# 6 Discussion and Conclusion

It has long been recognized that the model of perfect rationality does not adequately capture human decision-making because it neglects the numerous resource limitations that prevent the selection of the optimal choice [13]. In this work, we considered a model of bounded-rational decision-making inspired by ideas from statistical mechanics and information-theory. A distinctive feature of this model is the interplay between the decision-maker's preferences, a prior distribution over choices, and a resource parameter. To test the model, we conducted an experiment in which participants had to solve puzzles under time pressure. The experimental results are very well predicted by the model, which allows us to draw the following conclusions:

1. *Prior*: When the decision-making resources decrease, people's choices fall back on a prior distribution. This conclusion is supported by two observations. First, the bounded-rational model explains the gradual shift of the subjects' choice probabilities towards the prior as the duration of the trial is reduced (e.g. Fig.2). Second, the model fit obtained by the Softmax rule (13), which differs from the bounded rational model (7) only by the lack of a prior distribution, has a significantly larger error. Thus, our results conflict with the predictions made by models that lack a prior choice distribution—most notably with expected utility theory [11, 17] and the choice models based on the Softmax function (typical in reinforcement learning, but also in e.g. the *logit rule* of *quantal response equilibria* [5] or in *maximum entropy inverse reinforcement learning* [18]).

2. *Utility and Inverse Temperature*: Posterior choice probabilities can be meaningfully parameterized in terms of utilities (which capture the decision-maker's preferences) and inverse temperatures (which encode resource constraints). This is evidenced by the quality of the fit and the cogent operational role of the parameters. Utilities are stimulus-contingent enhancers/inhibitors that act upon the prior choice probabilities, consistent with the role of utility as a measure of relative desirability in *regret theory* [3] and also related to the cognitive functions attributed to the *dorsal anterior cingulate cortex* [12]. On the other hand, the inverse temperature captures a determinant factor of choice behavior that is independent of the preferences—mathematically embodied in the low-rank assumption of the log-likelihood function that we used for the decomposition in the analysis. This assumption does not comply with the necessary conditions for rational meta-reasoning, wherein decision-makers can utilize the knowledge about their own resources in their strategy [7].

3. *Preference Learning*: Utilities are learned from experience. As is seen in the utility functions of Fig. 3, subjects did not learn the optimal choice of the untrained stimulus (i.e. $x = 7$) in spite of being just a simple color-inversion of the most frequent stimulus (i.e. $x = 2$). Our experiment did not address the mechanisms that underlie the acquisition of preferences. However, given that the information necessary to establish a link between the stimulus and the optimal choice is below two bits (that is, far below the $\binom{3}{2} \cdot 2^2 \cdot 6 = 72$ bits necessary to represent an arbitrary member of the considered class of puzzles), it is likely that the training phase had subjects synthesize perceptual features that allowed them to efficiently identify the optimal solution. Other avenues are explored in [14, 15] and references therein.

4. *Diminishing returns*: The decision-maker's performance is marginally diminishing in the amount of resources. This is seen in the concavity of the expected utility curve (Fig. 4; similarly in the percentage of correct choices) combined with the sub-linear growth of the inverse temperature as a function of the duration (Fig. 3). For most subjects, the model predicts a perfectly-rational choice behavior in the limit of unbounded trial duration.

In summary, in this work we have shown empirically that the model of bounded rationality provides an adequate explanatory framework for resource-constrained decision-making in humans. Using a challenging cognitive task in which we could control the time available to arrive at a choice, we have shown that human decision-making can be explained in terms of a trade-off between the gains of maximizing subjective utilities and the losses due to the deviation from a prior choice distribution.

### Acknowledgements

This work was supported by the Office of Naval Research (Grant N000141110744) and the University of Pennsylvania.

## Footnotes

[1]For simplicity, here we consider only strictly positive values for the inverse temperature $\beta$, but its domain can be extended to negative values to model other effects, e.g. risk-sensitive estimation [9].

[2]More precisely, the 2-SAT and SAT problems are NL- and NP-complete respectively. This means that every other decision problem within the same complexity class can be reduced (i.e. rephrased) as a SAT problem.

[3]More precisely, $P(x, r, y) \propto N(x, r, y) + 1$, where $N(x, r, y)$ is the count of ocurrences of $(x, r, y)$.

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
