[Reviews · NeurIPS 2016]

Reviewer 1

Summary

This paper proposes a model of bounded-rational decision-making using ideas from statistical mechanics and information-theory, in particular assuming that humans adopt a choice policy which maximizes expected subjective utility while minimizing KL divergence from a prior choice distribution, with the trade-off determined by a temperature parameter that is related to the amount of time/computational resources available to the individual for deliberation. The model is tested on a set of data obtained from human subjects solving a pictorial version of the conunctive normal form (CNF) problem, and finds, based on the results, that subjects' prior choice distributions are similar to the true distribution over solutions, that the estimated temperature parameter varies monotonically with deliberation time allowed, and the estimated subjective utility peaks at the correct solution for well-trained puzzles but rather flat for an untrained puzzle.

Qualitative Assessment

Overall, this seems to be an interesting and mathematically sophisticated piece of work, bringing together quantitative tools from diverse feeds to bear on the modeling of bounded-rational decision making. I feel the conclusions/framing of the paper is somewhat over-reaching, because the model of choice (primarily Eq. 1) is not really a model of decision-making, nor of learning. What I mean is that the framework allows one to infer an individual's "subjective utility" (which from a normative perspective should be the distribution of correct answers for the trained puzzles), but does not say anything about how those utility preferences get computed, nor explain how these computations are learned or improved over time as the subject gets more experience with a particular puzzle. The application to the CNF pictorial puzzle data set seems original, but with the way the task is designed (often very short viewing time), the task really seems more like a memory task than a decision-making task. The estimated utility function being flat for an untrained puzzle makes it clear that nontrivial computations & learning are indeed involved. In fact, it's clear from looking at Eq. 1, that if a subject really had access to the utility function Ux(y), then he should just pick the maximal utility response every single time for a given stimulus x, as that would yield better empirical utility (accuracy rate) than maximizing the full Eq. 1. So Eq. 1 cannot be taken very literally as a model of how subjects make choices on each given trial, but rather is a descriptive model of how the distribution of choices might arise over a set of stimuli and presentations. The authors' feedback on this point did not assuage my concerns. Also, in terms of related work, there was a previous NIPS paper (Frazier & Yu, 2008) entitled "Sequential hypothesis testing under stochastic deadlines" that treated the problem of human decision-making under time pressure. While that took a very different approach at multiple levels, it is surely worth including as a reference in discussing how the current work situates in the ecology of human decision-making modeling literature. Unfortunately, the actual findings are not very surprising/interesting at a scientific level. That the inferred "subjective utility" should peak at the correct solution for learned puzzles is hardly surprising, if subjects are indeed capable of learning to do this task through repeated trials. However, it does serve as a useful method validation step, to establish for a problem where the "correct" utility is known, that the utility function+beta+prior estimation procedure indeed performs as desired. I would have liked to see an alternative model being compared to. In general it is good to have a baseline model as comparison so as to give some additional insight to the mechanism, correctness, and generality of the proposed model. In this particular case, because the way the decision-making model set up is so unusual compared to conventional practice, I think the readers would especially benefit from a comparison to (perhaps simpler) alternatives. The authors' feedback on this point did make me feel more reassured. However, the 80% subject dropout rate mentioned in the rebuttal is extremely high! This would suggest that perhaps the experimental procedure resulted in substantial "subject filtering" to get a good model fit, instead of taking the more sensible "model filtering." AIC was better for the B-R model in 21 out of 30 included subjects, but really it could have actually been only out of a total 120 subjects. That hardly makes this seem like a persuasive model of human decision-making. My final scores reflect the appearance/persistence of the various concerns that crystalized during the rebuttal phase. Other comments: - Sec. 2 is rather dense with math, and it is not always clear whether a mathematical statement is an assumption, a derivation, an explanation, or an interesting aside.

Confidence in this Review

2-Confident (read it all; understood it all reasonably well)


Reviewer 2

Summary

This paper presents a computational model of how humans make complex decisions with time constraints based on bounded rationality. The model assumes that people begin with a prior based on experienced outcomes, and with enough time can move toward the optimal solution for that decision. The paper presented a novel behavioural task and used the model to disentangle two components which contribute to decision-making: a utility function over possible solutions and a temperature parameter which controlled the trade-off between the prior utilities and optimal solution.

Qualitative Assessment

Overall, I thought this was a well-developed and really intriguing paper with lots of good ideas. My primary concern is really with the claim that this is a model of preferences and decision-making. The core task seems to be a puzzle-solving task, which, at best, is a stretch to compare to usual economic decision-making task. Some issues that could be improved/addressed: 1. With such a long exp’t on mTurk, there must have been many drop-outs. What was the drop-out rate in the experiment? Does this self-selection matter in terms of the quality of the model for fitting the general population (and not only those with the wherewithal to persist in answering puzzles like this on the computer for 50-130 minutes). 2. The bottom row of Figure 2 seems off. Was that overlay supposed to be there? If so, it needs to better explained. 3. Did the softmax model have the same numbers of parameters (I think so), but it would be good to be explicit to better justify the model comparison. Was there any attempt to do any out-of-sample predictions? 4. Figure 4 makes it seems that all the curves were fit to only 3 data points each. Those three points lie on the curve in each case, but the rest of the curve seems unsupported by the data. Perhaps more explanation of the way to interpret that visualization is in order. 5. The reason for the inclusion of the mutual information measure was not exactly clear? What was that supposed to add to the analysis? 6. The discussion/final paragraphs could use to be more prose rather than numbered bullet points.

Confidence in this Review

2-Confident (read it all; understood it all reasonably well)


Reviewer 3

Summary

The paper applies a model of bounded-rationality to data collected on human participants and try notably to account for the effect of time pressure.

Qualitative Assessment

I don't believe my opinion on that paper should be taken into account. Though it looks impressive and well-written to me, I do not believe I have the sufficient mathematical background to provide a fair assessment of its content.

Confidence in this Review

1-Less confident (might not have understood significant parts)


Reviewer 4

Summary

The authors proposed a decision-making model that takes into account time constraint. Traditional decision-making model is based on expected utility function. The authors added a time-weighted regularization term that is linked to one's prior knowledge. Therefore, under time constraint, one would weight more on her prior knowledge to make fast decisions. The authors used a combinatorial task with different time constraints to demonstrate the fitness of their model with experimental data.

Qualitative Assessment

The authors proposed a new model for time constrained decision making. To demonstrate that the new model is better, I'd like to see the authors quantitatively compare their model with previous models. Also the authors claim that there is a good match between the experimental data and the model. So there should be some kind of measurement to quantify the fitness between the data and the model ( and perhaps compared to old models) Last but not least, a model should be able to make predictions. The authors used all the data to fit the model, and 3 data points on a fitted nonlinear curve (Fig.4) is not a strong argument about fitness. What if the authors do a cross-validation? Would the model trained on part of the data be able to predict rest of the data points? If new experiments also fit on this curve that is based on current data, it would be more convincing.

Confidence in this Review

2-Confident (read it all; understood it all reasonably well)


Reviewer 5

Summary

PaperID 65, Human Decision-Making under Limited Time, presents a quantitative model of time-constrained decision-making based on expected utility and KL-divergence minimization. This model of bounded-rational decision making is evaluated using experimental human choice data in which subjects have to solve a sequence under time pressure. The experimenters designed the task to be performed under decreasing time limits to explore the effect of time constraints on decision-making. The expected utility function is defined as the free energy functional, where the posterior for a choice given a stimuli is is the maximizer over the free energy functional. Overall, the objective function's value is based on a trade off between minimizing information cost and maximizing expected utility. The free energy functional is proportional to the subjective utility of a choice and the inverse temperature which is a measure of the resources available for deliberation. The authors then generalize the equation into the form of the Gibbs distribution. A single parameter, beta, encapsulates constraints - where higher values approach perfectly rational behavior. Human data was collected on a choice experiment with limited time resources. Subjects were tasked with solving puzzles consisting of a Boolean formula in conjunctive normal form. Subjects picked colors in the pattern to satisfy a formula for truth assignment. The difficulty of the puzzles was fixed across trials. The experiments analyzed only the test trials. The formulation of the model fit well with human data and provided several insights. First, individuals rely on prior information when cognitive resources are challenged. Secondly, posterior choice probabilities can be modeled as a quantified metric of a decision-maker's preferences and current resource constraints. Third, utilities associated with choices are plastic and evolve with experience. Lastly, performance diminishes proportionally to amount of available resources.

Qualitative Assessment

Overall, the paper clearly outlines the problem and proposes a well defined mathematical framework for approaching and solving the problem. The use of experimental human data to validate the predictions of the model greatly increases the technical quality and impact of the proposed theoretical framework. Furthermore, the authors make much-needed links between the mathematical statistics and statistical mechanics literature with neuroscience. The results presented in this paper provide a foundation for neuroscientists who seek to develop computational models using behavioral and neuroimaging data. The mathematical prose is very well written and portrays the material as very approachable to a mathematically savvy newcomer.

Confidence in this Review

1-Less confident (might not have understood significant parts)


Reviewer 6

Summary

A bounded rational decision-making model based on the free energy formalism is fitted to human choice data under varying time constraints. The results indicate that, as computational resources (here, time) are reduced, human decision-making increasingly relies on stimulus-independent prior choice policies thus providing a normative account of some sub-optimal decision-making within the bounded rationality framework.

Qualitative Assessment

Overall, I believe this abstract would make a reasonably interesting contribution however I think (1) it is limited in scope in some respects thus reducing its potential impact, (2) an important technical issue must be addressed regarding the model comparison between the bounded rational model and the softmax model, and (3) some aspects of the experimental design could be clarified further and motivated better. (1) The model is very much phenomological in nature and does not examine neural or psychological mechanisms which, upon application to the task at hand, might reveal principles regarding how decision strategies arise in the face of complex problems and/or limited resources. The bounded rational model essentially proposes that there is a spectrum spanning from the prior choice distribution to the optimal choice distribution on each trial (as reflected in the utility functions) and time limits impede the decision-maker's progress along this spectrum to the correct choice (as quantified by the inverse temperature). This picture is agnostic to the algorithm that the subject's might be using to reason about the problems and generate decisions. (2) It appears to me that the bounded rational model has many more parameters being fitted (7, prior choice distribution) compared to the softmax model. Can the authors confirm or reject this? If confirmed, a correction for the difference in the number of fitted parameters should be incorporated when comparing error rates across models. (3) For example, in the shortest time window condition, after 0.25s of stimulus presentation, the subject is told that they have 1s to respond. This interruption suggests to me that subjects might tend to cease processing after 250ms, which is probably not enough time to even fully register the stimulus, and switch to a stochastic choice strategy. This is not inconsistent with the overall theme of the study but seems a somewhat trivial algorithmic "meta-effect" which I wonder if the authors can rule out? More generally, I think that more insight into the task design, its motivation, and how it interacts with possible decision mechanisms would be useful. Also, I'm unconvinced that choosing stimuli with a single solution is a good way to enforce a similar level of problem complexity since e.g. all black circles has only one solution but is the easiest problem. The authors could expand a bit on the conjunctive normal form and explain the notation in their boolean formula in fig 1d. Furthermore, the nontrivial prior distribution of stimuli should be discussed in detail since this defines the choice prior.

Confidence in this Review

3-Expert (read the paper in detail, know the area, quite certain of my opinion)